

# Transcriptional similarity in couples reveals the impact of shared environment and lifestyle on gene regulation through modified cytosines

Ke Tang[2] and Wei Zhang[1,3]

[1] Institute of Precision Medicine, Jining Medical University, Jining, China
[2] Department of Bioengineering, University of Illinois at Chicago, Chicago, IL, United States
[3] Department of Preventive Medicine, Northwestern University Feinberg School of Medicine, Chicago, IL, United States

Corresponding author
Wei Zhang,
weizhang.chicago@gmail.com

## ABSTRACT

Gene expression is a complex and quantitative trait that is influenced by both genetic and non-genetic regulators including environmental factors. Evaluating the contribution of environment to gene expression regulation and identifying which genes are more likely to be influenced by environmental factors are important for understanding human complex traits. We hypothesize that by living together as couples, there can be commonly co-regulated genes that may reflect the shared living environment (e.g., diet, indoor air pollutants, behavioral lifestyle). The lymphoblastoid cell lines (LCLs) derived from unrelated couples of African ancestry (YRI, Yoruba people from Ibadan, Nigeria) from the International HapMap Project provided a unique model for us to characterize gene expression pattern in couples by comparing gene expression levels between husbands and wives. Strikingly, 778 genes were found to show much smaller variances in couples than random pairs of individuals at a false discovery rate (FDR) of 5%. Since genetic variation between unrelated family members in a general population is expected to be the same assuming a random-mating society, non-genetic factors (e.g., epigenetic systems) are more likely to be the mediators for the observed transcriptional similarity in couples. We thus evaluated the contribution of modified cytosines to those genes showing transcriptional similarity in couples as well as the relationships these CpG sites with other gene regulatory elements, such as transcription factor binding sites (TFBS). Our findings suggested that transcriptional similarity in couples likely reflected shared common environment partially mediated through cytosine modifications.

## INTRODUCTION

Gene expression is a complex quantitative trait that may be influenced by both genetic and non-genetic factors, such as environment. Besides genetic variants identified as eQTL (expression quantitative trait loci), more recently, contributions from epigenetic systems, such as microRNAs, modified cytosines, and histone modifications to gene expression phenotypes have been investigated in various studies including those using
the International HapMap Project (*Hapmap, 2003*; *Hapmap, 2005*) lymphoblastoid cell lines (LCLs) derived from apparently healthy individuals (*Huang et al., 2011*; *McVicker et al., 2013*; *Moen et al., 2013*; *Zhang et al., 2014*). In addition, substantial gene-environment interactions have begun to be demonstrated in gene expression studies including a recent study based on transcriptomic sequencing assay in twins (*Buil et al., 2015*). Characterizing which genes are more likely to be influenced by environmental factors (e.g., shared living environment, behavioral lifestyle) as well as their relationships with genetic and epigenetic variations can enhance our understanding of human complex traits, given the fundamental roles of gene expression in determining traits and phenotypes.

Specifically, in this work, we utilized the unrelated couples from the HapMap YRI (Yoruba people from Ibadan, Nigeria) LCL panel to characterize gene expression pattern in couples, taking advantage of the whole-genome gene expression data that have been profiled from our previous publication using the Affymetrix Human Exon 1.0 ST Array (exon array) (*Zhang et al., 2008*). Genes showing the 'couple effect' of regulation (i.e., transcriptional similarity between husbands and wives) may reflect the shared common living environment and behavioral lifestyle in couples, who should be genetically independent in a random-mating society. To further evaluate the potential contribution by modified cytosines (i.e., methylation at CpG dinucleotides) to the 'couple effect' of regulation, we integrated our previously published cytosine modification data on these samples using the Illumina HumanMethylation450 BeadChip (450K array) (*Moen et al., 2013*). This work aims to shed novel light into clinical observations on spousal correlations of lifestyle-related risk factors for diseases (e.g., cardiovascular diseases) (*Jurj et al., 2006*; *Di Castelnuovo et al., 2009*), as well as implicate cytosine modifications as a critical epigenetic gene regulation mechanism that may mediate the shared environment and lifestyle.

## MATERIALS AND METHODS

The workflow, data analysis and thresholds used are summarized in Fig. 1.

### Detection of genes showing the 'couple effect' of regulation

Whole-genome gene expression data (GSE7851) on a collection of HapMap YRI LCL samples were previously generated using the Affymetrix Human Exon 1.0ST Array (exon array) (*Zhang et al., 2008*) . Sample preparation, array profiling, data processing, summarization, and normalization were described in our previous publication (*Zhang et al. 2008*). Selected genes from the exon array data have been experimentally validated (*Zhang et al., 2008*; *Zhang et al., 2009*). In total, 14,591 gene-level transcript clusters that were mapped to unique Entrez Gene IDs in 29 unrelated YRI couples (58 individuals) were used for testing the 'couple effect' of gene regulation. The 'couple effect' was measured by the difference of gene expression levels in a couple of sample *A* and sample *B*:

$$d_i^{A,B} = \frac{\left| X_i^A - X_i^B \right|}{\tilde{X}_i}$$

where $X_i^A$ and $X_i^B$ are the expression levels of gene $i$ in sample $A$ and $B$, separately. $\tilde{X}_i$ is the median of expression for gene $i$. To control false discovery rate (FDR), we calculated

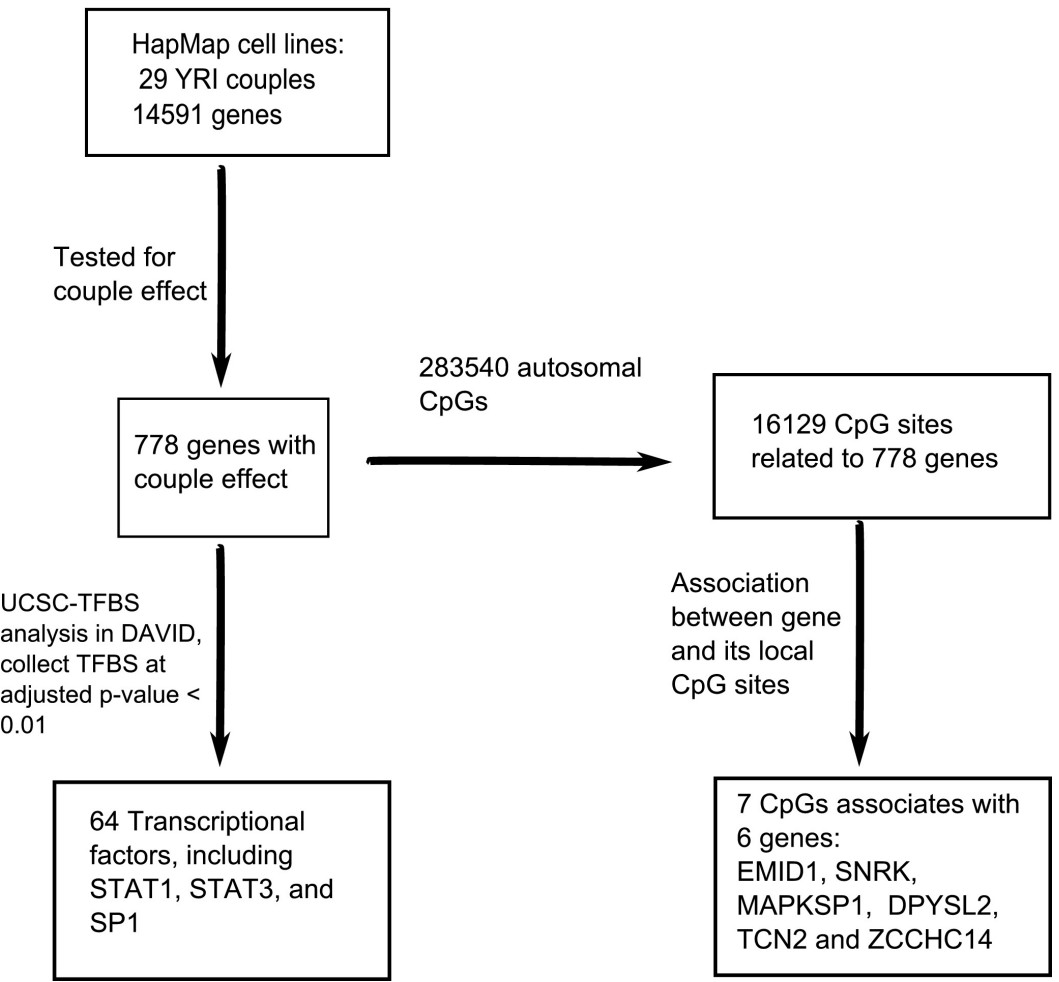

**Figure 1** **An overview of the workflow.** In total, 14,591 genes were tested for the couple effect in 29 Yoruba couples. 778 genes were detected at empirical $p$-value < 0.05. NIH/DAVID functional analysis showed 64 TFBS (transcription factor binding sites) enriched among these genes at FDR < 0.01. Among the total 283,540 autosomal CpGs, 16,129 CpGs are annotated to the 778 genes. Seven *cis*-acting associations were found between these genes and their local CpGs.

empirical $p$-values by performed 10,000 random samplings in these samples. In each sampling, we randomly assembled 29 pairs of males and females from the 58 individuals without replacement. The order of samplings was retained for all of the tested genes. For each gene, a one-tailed $t$-test was used to determine if the difference of gene expression in couples was smaller than that in the same number of random pairs. We also calculated Pearson correlations of gene expression differences between real couples and randomly assembled non-family male–female pairs.

## Linking modified cytosines to genes with the 'couple effect' of regulation

Cytosine modification data (GSE39672) were previously profiled by us using the Illumina Infinium HumanMethylation450 BeadChip platform (450K array) (*Moen et al., 2013*).

DNA sample preparation, 450K array profiling, data processing, summarization and normalization were described in our previous publication (*Moen et al.*, *2013*). Selected CpGs from the cytosine modification dataset have been experimentally validated using bisulfite sequencing (*Moen et al.*, *2013*). A total of 283,540 autosomal CpG sites *Zhang, Mu & Zhang* (*2012*) after removing CpG sites ambiguously mapped to the human genome and CpG sites containing common SNPs were included in the current study. We tested for correlation between gene expression levels and the $M$-values (*Du et al.*, *2010*) of local CpG probes, defined as CpG sites located within the 10 kb regions upstream of the transcription start sites (TSS) or downstream of the transcription end sites (TES) based on the RefSeq database hg19 (*Pruitt et al.*, *2014*).

### Functional annotation analysis

We used the DAVID (Database for Annotation, Visualization and Integrated Discovery) tool (*Huang Da, Sherman & Lempicki* , *2009b*; *Huang Da, Sherman & Lempicki* , *2009a*) to systematically search if there were any functional terms, such as Gene Ontology (GO) (*Ashburner et al.*, *2000*) biological processes and motifs (e.g., TFBS, transcription factor binding sites) enriched among the identified genes with the 'couple effect' of regulation relative to the human genome reference at a Benjamini adjusted $p$-value of 1%.

## RESULTS

### Transcriptional similarity in couples

Overall, based on the 14,591 analyzed genes, the correlations between male and female samples in couples are higher than non-family male–female pairs in the YRI samples (Fig. 2A). The peak of the correlation curve of true couples tended to shift towards the higher correlation values compared to non-family male–female pairs. In total, 778 gene-level transcript clusters showed significantly smaller variance couples than in random pairs of males and females at an empirical $p$-value < 0.05 (Table S1). The distribution of empirical $p$-values (Fig. 2B) showed an obvious bias towards genes with smaller variance in couples. For these 778 genes with the 'couple effect' of regulation, the correlations between male and female samples in true couples were significantly higher than non-family male–female pairs (Fig. 2C). The mean and median of correlations in true couples were 0.984 and 0.986, versus 0.971 and 0.972 in non-family male–female pairs. The overlapped area under the curve (AUC) is 0.22, indicating that gene expression levels of a substantial number of genes in couples are more similar compared to random pairs of individuals.

### Linking modified cytosines with genes showing the 'couple effect' of regulation

In total, 16,129 local CpG sites (i.e., within 10 kb up- and down-stream of genes) were annotated to the 778 genes with the 'couple effect' of regulation. These CpG sites can be grouped into three major categories: upstream (−10 kb to TSS) (3,404 CpGs), gene body (9,261 CpGs), and downstream (TES to +10 kb) (3,464 CpGs). At $q$-value < 0.05, seven local CpG sites were found to be associated with six genes in the YRI samples: *EMID1* (encoding EMI domain containing 1), *SNRK* (encoding SNF related kinase),

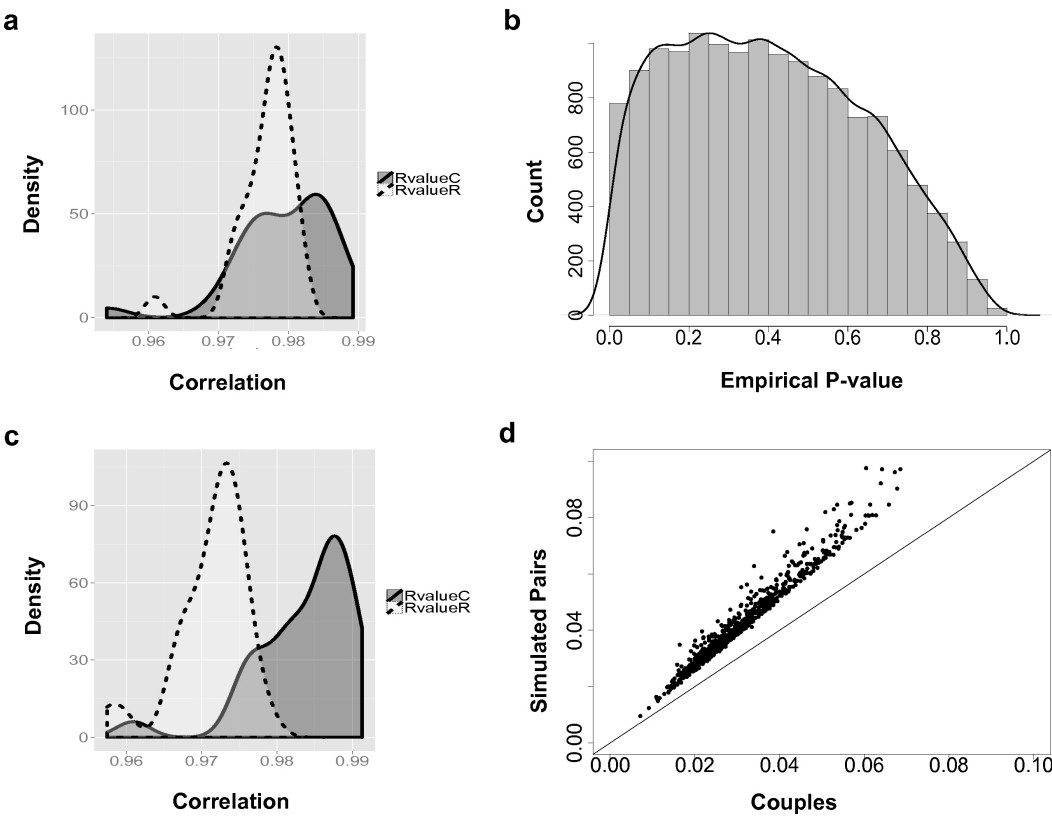

**Figure 2** **The couple effect of genetic expression.** (A) The density plot of correlations between male and female samples in true couples (RvalueC) and non-family male-female pairs (RvalueR) in all 14,591 genes. (B) The distribution of empirical *p*-values of couple effect of gene expression. (C) The density plot of correlations between male and female samples in true couples (RvalueC) and non-family male-female pairs (RvalueR) in the 778 genes at empirical *p*-values < 0.05. (D) Differences between male and female samples in gene expression. For genes with couple effect detected at 5% empirical *p*-value, the average differences between male and female samples from simulated (*Y*-axis) were compared with the male–female differences from true couples (*X*-axis).

*MAPKSP1* (encoding MAPK Scaffold Protein 1), *DPYSL2* (dihydropyrimidinase-like 2), *TCN2* (encoding transcobalamin II) and *ZCCHC14* (encoding zinc finger, CCHC domain containing 14) (Table 1). For example, the modification levels of CpG site cg24811472, located in the gene body, were positively associated with the expression levels of *DPYSL2* (Fig. 3A). Among these six CpG-regulating genes, *DPYSL2* is known to be associated with schizophrenia and bipolar disorder (*Fallin et al.*, *2005*); while *TCN2* is associated with various disorders including Alzeimer's disease, vascular disease, and certain cancers (e.g., brain and colorectal) (*Hazra et al.*, *2010*).

## Functional annotation analysis

We searched for enriched functional annotations among the 778 genes with the 'couple effect' of regulation in the YRI samples. Notably, 64 TFBS (Table S2) were detected to be associated with these genes at a Benjamini adjusted *p*-value < 1%. Among them, the exon array data of 29 transcription factors (TF) are available for testing associations between

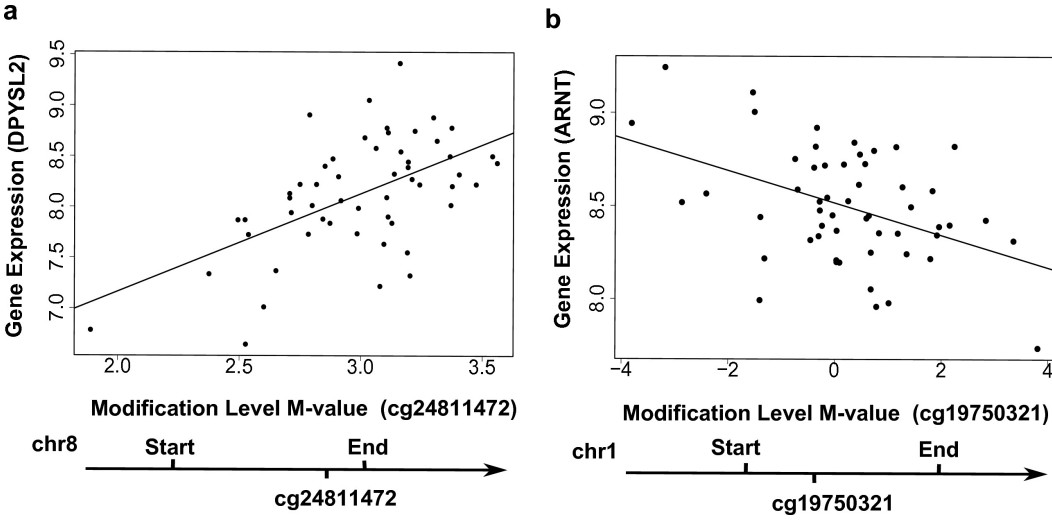

**Figure 3** **Examples showing that cytosine modifications account for the 'couple effect' of gene expression.** (A) Gene expression level of *DPYSL2* (encoding dihydropyrimidinase-like 2) is significantly associated with a gene-body CpG site cg24811472 ($p = 1.06E{-}05$). (B) Gene expression level of *ARNT* (encoding aryl hydrocarbon receptor nuclear translocator) is significantly associated with a gene-body CpG site cg19750321 ($p = 0.001$).

**Table 1** **CpG-Gene expression associations at *q*-value < 0.05 between genes with the 'couple effect' and their local CpGs.**

| CpGID | Gene Symbol | Affymetrix ID | CpGvalue | *p*-value | *q*-value | r |
|---|---|---|---|---|---|---|
| cg02152034 | EMID1 | 3941848 | −0.3732414 | 1.42E–06 | 0.0227167 | −0.602693 |
| cg10527635 | SNRK | 2619666 | −0.4224054 | 6.71E–06 | 0.04227625 | −0.570362 |
| cg12001078 | MAPKSP1 | 2779408 | −0.2527613 | 8.37E–06 | 0.04227625 | −0.565482 |
| cg24811472 | DPYSL2 | 3091077 | 0.9650537 | 1.06E–05 | 0.04227625 | 0.560246 |
| cg17759595 | DPYSL2 | 3091077 | 0.5309326 | 1.33E–05 | 0.04262027 | 0.554953 |
| cg04081402 | TCN2 | 3942472 | −0.5538604 | 1.70E–05 | 0.04535223 | −0.549258 |
| cg06545761 | ZCCHC14 | 3703665 | 0.0759574 | 1.99E–05 | 0.04554208 | 0.545513 |

local CpGs and gene expression of TF. A number of local CpGs were significantly associated with the TF gene expression at $p < 0.05$ (Table S3). For instance, the modification levels of CpG site cg19750321 were negatively associated with *ARNT* (encoding aryl hydrocarbon receptor nuclear translocator) expression in the YRI samples (Fig. 3B). These results indicated a potential route of regulation, in which local TF CpGs may be influenced by environment; TF CpGs may regulate their corresponding TF gene expression; then TF regulate target gene expression. In addition, 19 out the 778 genes are located on chromosome 19p13.3, representing an enrichment of 4.2-fold relative to the human genome reference (Bemjamini adjusted $p = 4.5E{-}4$). Interestingly, *LDLR* (encoding low density lipoprotein receptor), located on chromosome 19p13.3 is known to be associated with various diseases, such as coronary artery disease and dyslipidemia (*Martinelli et al., 2010*). Couple concordance has been found in coronary artery disease. The total serum cholesterol and low-density lipoprotein (LDL) cholesterol were significantly lower for

the wives whose husbands exposed to a continuous coronary heart disease risk-factor intervention program (*Sexton et al.*, *1987*).

## DISCUSSION

In this paper, our findings demonstrated that a substantial number of genes showed smaller variances in couples than random pairs of males and females. A link between cytosine modifications to some of these genes with the 'couple effect' of regulation suggested that modified cytosines contributed partially to this gene expression pattern in couples, thus likely mediating the influence of shared living environment and behavioral lifestyle on gene expression among family members.

Elucidating environmental effect on gene expression regulation will enhance our understanding of complex traits and diseases. Clinically, spousal concurrences of certain diseases are not uncommon. For example, people are found at increased risk of having asthma, hypertension, hyperlipidaemia and pepticulcer diseases when their spouses have these diseases (*Hippisley-Cox et al.*, *2002*). In this study, taking advantage of the HapMap LCL samples, on which we have accumulated a tremendous resource of gene expression and epigenomic data (*Zhang, Zheng & Hou*, *2013*), we explored to identify a substantial number of genes showing the 'couple effect' of regulation (i.e., transcriptional similarity in couples) in the YRI samples derived from African individuals. Though overall, there was no significant enrichment of GO biological processes or canonical pathways among the 778 genes with the 'couple effect' of regulation, some of these genes are known to be associated with certain traits and diseases that have been shown to be familial. Notably, chromosome 19p13.3, which contains 19 genes showing the 'couple effect' of regulation, was enriched relative to the human genome. Genes on chromome 19p13.3 have been associated with various diseases. In particular, *LDLR*, a gene showing the 'couple effect' of regulation in this study, has been associated with familial hypercholesterolemia and coronary artery disease (*Martinelli et al.*, *2010*). In a previous study, couple concordance was found in coronary artery disease as the total low-density lipoprotein (LDL) cholesterol and serum cholesterol were significantly lower for the wives whose husbands exposed to a continuous coronary heart disease risk-factor intervention program compared to the wives from control group (*Sexton et al.*, *1987*). Our findings, therefore, suggested a potential link between shared living environment and behavioral lifestyle and the expression patterns of a number of genes, which in turn may explain spousal or familial concurrences of certain diseases and phenotypes.

We further evaluated whether epigenetic systems, specifically cytosine modifications might contribute to the observed 'couple effect' of regulation. Modification levels of local CpG sites were found to account for several genes that showed the 'couple effect' of regulation. Among these CpG-regulated genes (Table 1), a few are known to be associated with complex diseases. For example, two local CpG sites in body regions were associated with the expression levels of *DPYSL2*, which is known to be associated with psychiatric disorders including schizophrenia and bipolar disorder. In another example, one local CpG site in body region was associated with the expression of *TCN2*, which is associated

with Alzeimer's disease and certain cancers including colorectal cancer (*Hazra et al.*, *2010*). Interestingly, our results indicated a possible link of shared living environment and behavioral lifestyle in the regulation of *TCN2* that belongs to the one-carbon metabolism pathway, which has been associated with the risk for colorectal cancer. Since in a random-mating society, genetic background in couples is independent, CpG modifications, therefore likely function as the mediators for the environmental effect on gene expression in a shared living environment. In addition to association of local CpG sites and some genes with the 'couple effect' of regulation, there were also a few cases in which CpG modifications mediating environmental factors through regulating transcription factors, which in turn may regulate the target genes that showed the 'couple effect' of regulation.

In this study, we used couples from the HapMap YRI panel to represent shared environment for a significant length of time (i.e., long enough to have children). It may not be as ideal as using a twin design to dissect environmental factors from genetic factors, but using these HapMap couple data allowed us to explore directly how epigenetic systems may mediate the shared living environment and behavioral lifestyle in regulating gene expression in a well-characterized collection of samples, on which both gene expression and cytosine modification data are available. Other potential limitations may include the lack of age information on the HapMap samples, given that age is a likely factor affecting DNA methylation. Our findings warrant future more comprehensive investigations that integrate other critical epigenetic systems such as histone modifications to elucidate the 'couple effect' of gene regulation, which could improve our understanding of the complex relationships between environmental factors (e.g., lifesyles, behavior, air pollution, diet) and health conditions.

### Funding

This work was supported, in part by grants from the National Institutes of Health, HG006367 (to WZ). The funders had no role in study design, data collection and analysis, decision to publish, or preparation of the manuscript.

### Grant Disclosures

The following grant information was disclosed by the authors:
National Institutes of Health: HG006367.

### Competing Interests

The authors declare there are no competing interests.

### Author Contributions

- Ke Tang and Wei Zhang conceived and designed the experiments, performed the experiments, analyzed the data, contributed reagents/materials/analysis tools, wrote the paper, prepared figures and/or tables, reviewed drafts of the paper.

## Data Availability

The raw data has been supplied as a Supplemental Dataset.

## Supplemental Information

Supplemental information for this article can be found online at http://dx.doi.org/10.7717/peerj.2123#supplemental-information.

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
