# Peer review of "Transcriptional similarity in couples reveals the impact of shared environment and lifestyle on gene regulation through modified cytosines"

_PeerJ, doi:10.7717/peerj.2123_

## Round 0.1 · original submission · Major Revisions

Both reviewers were generally positive about the work, but both identified areas which need additional data/clarification and/or description. Please see the reviews from the two reviewers for more specifics.

·

Basic reporting

This manuscript reached the standards of PeerJ

Experimental design

I am generally satisfied with the experimental design but I think more use could have been made of genome-wide methylation data. see also minor points in my comments to authors.

Validity of the findings

I am happy with the conclusions made bar the minor points I raise with the authors

Additional comments

This is a well-written manuscript with a generally simple message: couples who live together have more similar expression patterns, showing evidence for the effect of shared environment on gene expression as a phenotype. Although this was known from studies of twins and others, this manuscript is sufficiently original to be published. My main suggestion concerns the methylation dataset. In addition to analysing methylation in genes that are most influenced by shared environment, it would be useful to perform a genome-wide analysis on the same dataset and contrast the findings with gene expression. I would also like to see a summary table of ages, sex etc for all participants. Minor points are:
1. Can you rule out that couples came from the same tribe/geographical area i.e. the shared environments experiences before marriage could also have influenced their genomes?
2. Can you test your hypothesis of random mating by looking at the DNA sequence analysis? This should clear up the possibility that couples that are more similar for expression are more similar for DNA sequence.

Reviewer 2 ·

Basic reporting

The authors investigate the impact of “shared environment” on gene expression levels, using a clever experimental design that compares “couples” (husband and wife) against random male-female pairs. Overall, I think this paper is of interest to PeerJ’s readership.

The following analysis is not clear:
On page 6 (methods description), I am not clear on what is being correlated to what: “We also calculated Pearson correlations between couples and non-family male-female pairs”. Can you clarify this?

Experimental design

No comments

Validity of the findings

Authors don't validate/replicate their finding. This may be outside the scope of this paper, but would be super informative if the authors could find a comparable dataset to replicate some of the top genes.

As a minimum, the authors could replicate the reported correlation between methylation probes and gene expression levels.

Additional comments

- It would be informative to try to disentangle the contribution of various environmental factors. Age is a simple one that comes to mind (“real” couples are more likely to have similar ages). Can the authors comment on the contribution of shared age? (e.g., if age is modelled, do you still find many of the same significantly “conserved” genes?)

- Although I appreciate the correlation between “modified cytosines” and gene expression, I think the experiment can be done in a much more conclusive manner (in terms of a mediation analysis). Right now, the authors correlate expression levels with “M-levels”. But this doesn’t tell us if shared environmental effect is mediated through methylation. I see two simple ways of assessing this: formal mediation analysis, or investigating conservation methylation level for the probes nearby the 778 genes.

- Is the shared environment impact greater than the impact of gender? (e.g., authors can construct a “null” that involves couples of the same gender).

---

## Round 0.2 · accepted · Accept

Overall looks good. Be sure to make sure that there are no remaining typographical or other errors that have been missed.

·

Basic reporting

See "general comments" below

Experimental design

See "general comments" below

Validity of the findings

See "general comments" below

Additional comments

I am satisfied with all revisions in response to both reviewers